# Acceptability of bisphosphonates among patients, clinicians and managers: a systematic review and framework synthesis

Zoe Paskins,[1,2] Fay Crawford-Manning [iD],[1,2] Elizabeth Cottrell [iD],[1] Nadia Corp,[1] Jenny Wright,[1] Clare Jinks,[1] Simon Bishop,[3] Alison Doyle,[4] Terence Ong,[5] Neil Gittoes,[6] Jo Leonardi-Bee,[7] Tessa Langley,[8] Robert Horne,[9] Opinder Sahota[10]

For numbered affiliations see end of article.

**Correspondence to**
Dr Zoe Paskins;
z.paskins@keele.ac.uk

## ABSTRACT

**Objective** To explore the acceptability of different bisphosphonate regimens for the treatment of osteoporosis among patients, clinicians and managers, payers and academics.

**Design** A systematic review of primary qualitative studies. Seven databases were searched from inception to July 2019. Screening, data extraction and quality assessment of full-articles selected for inclusion were performed independently by two authors. A framework synthesis was applied to extracted data based on the theoretical framework of acceptability (TFA). The TFA includes seven domains relating to sense-making, emotions, opportunity costs, burden, perceived effectiveness, ethicality and self-efficacy. Confidence in synthesis findings was assessed.

**Setting** Any developed country healthcare setting.

**Participants** Patients, healthcare professionals, managers, payers and academics.

**Intervention** Experiences and views of oral and intravenous bisphosphonates.

**Results** Twenty-five studies were included, mostly describing perceptions of oral bisphosphonates. We identified, with high confidence, how patients and healthcare professionals make sense (coherence) of bisphosphonates by balancing perceptions of need against concerns, how uncertainty prevails about bisphosphonate perceived effectiveness and a number of individual and service factors that have potential to increase self-efficacy in recommending and adhering to bisphosphonates. We identified, with moderate confidence, that bisphosphonate taking induces concern, but has the potential to engender reassurance, and that both side effects and special instructions for taking oral bisphosphonates can result in treatment burden. Finally, we identified with low confidence that multimorbidity plays a role in people's perception of bisphosphonate acceptability.

**Conclusion** By using the lens of acceptability, our findings demonstrate with high confidence that a theoretically informed, whole-system approach is necessary to both understand and improve adherence. Clinicians and patients need supporting to understand the need for bisphosphonates, and clinicians need to clarify to patients what constitutes bisphosphonate treatment success. Further research is needed to explore perspectives of male patients and those with multimorbidity receiving

### Strengths and limitations of this study

► Comprehensive search strategy.
► Robust framework synthesis underpinned by theory.
► Inclusion of clinician and manager views in addition to patient perspectives.
► Use of Grades of Recommendation, Assessment, Development, and Evaluation Confidence in the Evidence from Qualitative Reviews to give confidence in findings.
► Qualitative studies reviewed for inclusion were frequently not specific about the anti-osteoporosis drugs participants were taking, meaning we may have missed papers or over-interpreted findings.

bisphosphonates, and patients receiving intravenous treatment.

**PROSPERO registration number** CRD42019143526.

## INTRODUCTION/BACKGROUND

Osteoporosis is a disease that is characterised by skeletal fragility and changes in bone microarchitecture resulting in increased risk of fractures with no or low trauma.[1] The management and care of people with low trauma or fragility fractures results in considerable societal economic burden, annual cost in the UK alone is £4.4 billion.[2] Furthermore, the personal impact of fragility fractures is considerable, with potential deleterious effects on physical and psychological health, ability to live independently and increased risk of death. Many of these fractures are potentially preventable with appropriate cost effective and clinically effective drug treatments such as bisphosphonates, the mainstay of osteoporosis treatment. However, the success of treatment depends on patients initiating (starting), executing (or implementing—taking correctly) and persisting (continuing) medication; collectively these

processes are described as adherence. Adherence with osteoporosis medications is notoriously poor and reported to be poorer than other disease areas. Oral bisphosphonate persistence rates at 1 year are commonly estimated between 16% and 60%.[3] Worldwide, many people who would benefit from osteoporosis drugs are not receiving them, and this treatment gap has been described as an 'osteoporosis crisis'.[4] The treatment gap is compounded by poor adherence which results in potentially preventable fragility fractures with their associated burden for patients and their carers, difficulties in professional–patient relationships, and wasted healthcare resources.[5]

There are a number of different bisphosphonates, some are administered orally, others intravenously. A variety of regimes in terms of dose frequency also exists. Alendronic acid, an oral once-weekly bisphosphonate, is considered first-line and most commonly used.[6] Bisphosphonates work to reduce fracture risk. A recent network meta-analysis demonstrated that bisphosphonate treatment reduces the risk of fragility fracture (depending on site) by 33%–54%.[7] Oesophageal or gastrointestinal related side effects are the most common adverse effects of oral bisphosphonate use. To counter these, patients taking oral bisphosphonates are required to remain upright and fast for half an hour after ingestion. Rare side effects of bisphosphonates include osteonecrosis of the jaw and atypical femur fractures, both of which have received significant media attention. Such media reports are temporally related to declining bisphosphonate use.[7] Due to the gastrointestinal side effects and special instructions for taking oral treatment, it has been suggested that alternative bisphosphonate regimens, for example, annual intravenous zoledronic acid, may promote long-term adherence.[8–11] Studies to date which have examined patient preferences for osteoporosis treatment, suggest that patients prefer injections given less frequently[12–14]; however, research in other chronic diseases shows that although adherence is improved with less frequent medications, that patients prefer oral to injection treatment.[15] In osteoporosis, the majority of studies that explore patient preferences employ quantitative methods, for example, discrete choice experiments, where patients are asked to choose between hypothetical treatments in regards to various attributes (eg, efficacy, side effects, route and frequency of administration).[13] Such studies cannot provide comprehensive insight into patient views, experiences or the explanations for these preferences.

In order to fully understand the osteoporosis treatment gap, and ultimately improve adherence, it is important to understand perspectives of all relevant stakeholders: patients, healthcare professionals (HCPs), managers, payors and academics.[16 17] This can be achieved using the lens of 'acceptability', defined as 'a multi-faceted construct that reflects the extent to which people delivering, or, receiving a healthcare intervention consider it to be appropriate, based on anticipated or experienced cognitive and emotional responses to the intervention'.[18 19] In the context of a research programme designed to determine the research agenda for optimising bisphosphonate treatment, the primary aim of this systematic review is to explore the acceptability of different bisphosphonates regimens among patients, and clinicians and managers.

## METHODS
We conducted a systematic review and framework synthesis of qualitative studies exploring patient and clinician views and experiences of bisphosphonates. The conduct and reporting of this review followed the Preferred Reporting Items for Systematic Reviews and Meta-Analyses (PRISMA) guidelines (see online supplemental file 1 for PRISMA checklist).

### Eligibility
To be eligible for inclusion, studies needed to report on patients', clinicians', academics' and/or manager/payers' experiences and preferences regarding bisphosphonate regimes for adults (≥18 years) with osteoporosis. Bisphosphonates needed to be mentioned by name, or there needed to be sufficient information that was specific to bisphosphonate (eg, reference to the special instructions for use of oral bisphosphonates), to deduce that study findings related to bisphosphonates, as agreed by two clinically experienced authors independently. Papers describing experiences of osteoporosis more generally were included if there were findings relating to bisphosphonate treatment in the study abstract. Studies were only included if they were qualitative in design, or mixed methods with a qualitative component, relevant to a developed country setting and written in English language. Studies were excluded that involved paediatric patients; patients and clinicians receiving/recommending other treatments for osteoporosis; and studies in which bisphosphonates were being used for other indications (eg, malignancy or Paget's disease).

### Search methods
Systematic searches were conducted in seven bibliographic databases (MEDLINE, EMBASE, AMED, CINAHLPlus, PsycINFO, ASSIA, and Web of Science (Social Science Citation Index and Conference Proceedings Citation Index-Social Science and Humanities)) from inception to 15 July 2019. The search strategy used database subject headings and text word searching in title, abstract or keywords, combining terms for: (1) bisphosphonates; (2) experiences and preferences; and (3) qualitative research, based on DeJean et al's search filter (see online supplemental file 2 for full MEDLINE search strategy).[19] Search terms were adapted as appropriate for each database platform.

In addition, grey literature was searched (DART Europe, Open Grey and National Digital Library of Theses and Dissertations); the reference lists of all included studies and relevant systematic reviews identified were checked and key studies were citation tracked.

## Study selection

Two-stage screening of articles against eligibility criteria was undertaken. First, titles and abstracts were screened, then full texts. At both stages screening was conducted by sets of two reviewers independently (NC, EC, ZP) and articles were excluded by agreement. Disagreements were resolved through discussion or by third reviewer adjudication.

## Data extraction

For each paper data extraction was completed independently by two researchers (ZP and JW or EC and FM). Key findings from the results sections of papers relating to bisphosphonates were extracted; a 'key finding' was defined as any sentence or statement relating to views or experiences of bisphosphonates from the results section of the paper or abstract. Wherever possible, the key finding was extracted as written by the author, with minimal edits only for clarification, description of context or for consistency across papers. For each paper, two authors extracted key findings independently, and subsequently agreed a final list of key findings for each paper. Data were also extracted on participant numbers and demographics, data collection technique, setting and country. Additionally, if available for patients, information was extracted on their bisphosphonate use including type of drug and current status (adherent, non-adherent, decliner).

## Quality appraisal

The quality of each study was assessed using the Critical Appraisal Skills Programme (CASP) qualitative tool. This tool consists of 10 items split into 3 sections (qualitative suitability, data analysis and overall quality) (online supplemental file 2). The first two sections consist of items related to qualitative suitability and data analysis, which were evaluated as 'yes', 'no', 'unclear' or 'partial'. The final question was an assessment based on the overall quality of the paper; this was informed by response to the previous items (indicating methodological quality) and by the relevance of the study to the review objectives and was rated as 'high', 'moderate' or 'low'. All papers were quality appraised by two researchers independently (FM, SB, JW). Disagreements were resolved through discussion with a fourth reviewer (ZP).

## Synthesis

We used a framework synthesis approach informed by the 'best fit' model described by Carroll *et al*.[20] The 'best fit' method offered a means to test, reinforce and build on an existing published model, conceived for a different but relevant purpose. This approach was chosen as a published theory was identified from the literature that conceptualised acceptability—the theoretical framework of acceptability (TFA).[18] The TFA is a relatively new framework which was developed to inform the understanding of acceptability of complex interventions, and consists of seven constructs: affective attitudes—the emotions elicited by an intervention; intervention coherence—the extent to which an intervention makes sense; perceived effectiveness—the perceived extent to which intervention will achieve purpose; burden—the amount of effort required to participate in an intervention; self-efficacy—individual's confidence that they can perform the behaviour(s) required to participate in the intervention; opportunity-costs—the extent to which benefits, profits, or values must be given up to engage in an intervention; and ethicality—the extent to which an intervention has a good fit with an individual's values. The framework also incorporates temporal perspectives on *anticipated* and *experienced* acceptability at three time points before (prospective), during (experienced) and after (retrospective) experience of an intervention.

The TFA has not previously been used to evaluate drug acceptability. We anticipated the seven constructs of the TFA would be relevant to engagement with drug treatment; for example, burden could relate to treatment burden associated with administrating the drug or side effects. However, one aspect which did not appear to be explicitly conceptualised within the framework was patient beliefs about medicines. Studies across a range of long term conditions, healthcare systems and cultures have consistently shown that engagement with treatment is influenced by patients' personal evaluation of the medicine in question.[21] Particularly important is how they judge their personal need for treatment relative to their concerns about it. For this reason, we therefore included the Necessity Concerns Framework (NCF),[21] to further explore the TFA domain relating to intervention coherence.

The first author initially conducted inductive open coding on the data extracted, before mapping the codes to a draft framework derived from a priori themes (the domains of the TFA). Authors then met to first discuss the themes and compare findings for each study and the 'fit' to the draft framework. A preliminary synthesis was achieved using tabulation of studies, organising the studies into groups relating to temporal perspectives and research question, and exploring relationships between studies and between groups.

A final coding framework was agreed at a second meeting of authors. A second author (FM) recoded the original key findings, where necessary, to the new framework to ensure all findings were represented. Finally, relationships between themes and TFA and NCF domains were explored by further group discussion. We used the Grades of Recommendation, Assessment, Development, and Evaluation Confidence in the Evidence from Qualitative Reviews (GRADE-CERQual) approach to determine confidence in our synthesised findings.[22]

## Patient and public involvement

Members of the Nottingham National, Royal Osteoporosis Society Support Group were involved in a series of meetings to discuss the design of the overarching research programme in which this study sits, and confirmed that

 

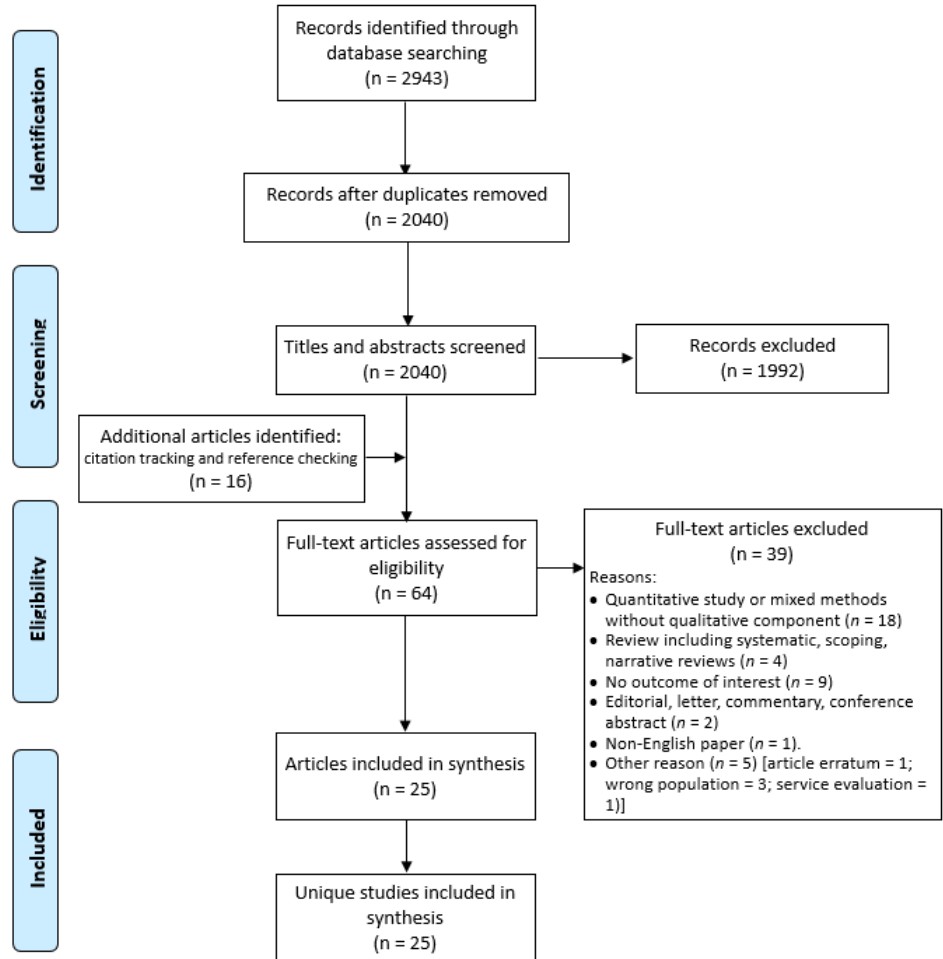

**Figure 1** Preferred Reporting Items for Systematic Reviews and Meta-Analyses diagram.

understanding acceptability of bisphosphonates from a range of perspectives was important. Patient were not directly involved in the conduct of this study.

## RESULTS

The literature search identified 2040 unique articles, of which 25 met eligibility criteria (figure 1), a summary of the studies is shown in table 1.

The included studies were categorised into three groups: perceptions of osteoporosis generally[23–29]; health-care service delivery issues unrelated to osteoporosis (de-prescribing[30] and inter-professional communication in primary care[31]) and studies specific to osteoporosis treatments. The latter group was further subdivided into: those examining treatment barriers[16 32–36]; adherence[37–39]; decision-making[40–44]; or bisphosphonate-related side effects.[45 46] Only one study examining adherence and one examining decision making had research questions which specifically related to bisphosphonates.[38 43]

The majority (23) of studies were conducted in North America or Europe. Eighteen studies explored patient views,[16 23–27 33 35 37–46] of which eight included men, and one study recruited patients taking anti-osteoporosis

drugs for glucocorticoid-induced osteoporosis.[36] Twelve studies explored HCPs' views,[16 28–32 34–36 39 42 43] and two studies interviewed managers.[16 34] No studies included academic or payor participants. Of the 18 studies that included patients, 10 studies described how many of the patients were on anti-osteoporotic medication, however, only two reported the specific type of medication. Only one study reporting patient experience of receiving intravenous bisphosphonate.[27]

The findings related to quality appraisal are summarised in table 2. The most common limitations of the included studies were lack of description of author reflexivity, lack of depth of analysis, use of normative statements and relatively small samples or studies conducted in a single site which may limit transferability of the findings. Furthermore, although the characteristics of the sample were generally reasonably described, in order to address our research question, we required information about medication use of participants which was frequently not described.

Using the CASP tool, 12 (48%) studies were scored as high value and the remaining 13 (52%) studies as moderate value. For 5/13 (38%) studies scored as

**Table 1** Summary of included studies

| Author | Participants | Participant no. (male:female) | Bisphosphonate use and adherence† | Data collection methods | Qualitative approach or analysis method‡ | Recruitment setting | Country |
|---|---|---|---|---|---|---|---|
| **Studies in group 1: views of osteoporosis** | | | | | | | |
| Besser et al[23] | Pts | 14 (0:14) | AOD unspecified | Interview | Framework analysis | One hospital | UK |
| Jaglal et al[29] | HCPs Family physicians | 32 (12:20) | N/A | Focus group | Constant comparison | Primary care | Canada |
| Otmar et al[28] | HCPs GP (n=14) Practice nurse (n=2) | 16 (11:5) | N/A | Focus group | Analytic comparison Constant comparison | Primary care | Australia |
| Sale et al[24] | Pts | 28 (2:26) | 19/28 pts on AOD adherent (n=19) declined (n=4) | Interview | Phenomenological study | National osteoporosis patient group | Canada |
| Sale et al[25] | Pts | 24 (6:18) | 9/24 pts on AOD, risedronate (n=8), etidronate (n=1) | Focus group | Descriptive qualitative study | Fracture clinic | Canada |
| Weston et al[26] | Pts | 10 (0:10) | AOD unspecified | Interview | Interpretative phenomenological analysis | Primary care | UK |
| Hansen et al[27] | Pts | 15 (0:15) | AOD unspecified adherent (n=12) declined/stopped AOD (n=3) | Interview | Phenomenological hermeneutic approach | Women attending DXA at 2 hospitals | Denmark |
| **Studies in group 2: views of osteoporosis treatment (treatment barriers)** | | | | | | | |
| Alami et al[35] | Mixed | Pts: 37 (0:37) HCPs: 18 (8:10) | 23/47 pts on AOD, adherent (n=19) declined/ stopped AOD (n=18) | Focus group | Grounded theory | Hospital/community over 5 regions | France |
| Drew et al[34] | HCPs Nurse (n=14), GP (n=2), Specialists (n=17), Orthopaedic surgeon (n=4) Managers (n=5) DXA technician (n=1) | 43 (not given) | N/A | Interview | Thematic approach | 11 hospitals in one region | UK |
| Feldstein et al[16] | Mixed | Pts: 10 (0:10) HCPs: 57 (not given) | AOD unspecified | Interview and focus group | Content analysis | Primary and secondary care | USA |
| Guzman-Clark et al[36] | HCPs | 23 (13:10) | 24/100 pts on AOD | Focus group | Thematic content analysis | Urban academic medical centre | USA |
| Merle et al[32] | HCPs (GP) | 16 (11:5) | N/A | Interview | Descriptive thematic analysis | Primary care | France |
| Merle et al[33] | Pts | 98 (53:45) | AOD Unspecified | Focus group | Inductive thematic analysis | Recruited from 2 existing research studies and community (medical insurance company) | France |
| **Studies in group 2: views of osteoporosis treatment (adherence)** | | | | | | | |

Continued

**Table 1** Continued

| Author | Participants | Participant no. (male:female) | Bisphosphonate use and adherence† | Data collection methods | Qualitative approach or analysis method‡ | Recruitment setting | Country |
|---|---|---|---|---|---|---|---|
| Iversen et al[39] | Mixed | Pts: 32 (2:30) HCPs: 12 (5:7) | AOD unspecified | Focus group | Open coding (thematic analysis) | Secondary care | USA |
| Lau et al[37] | Pts | 37 (0:37) | 33/37 pts on AOD, alendronate (n=9), etidronate (n=5), risedronate (n=19) | Focus group | Mixed phenomenological design | Primary care, secondary care and community pharmacies | Canada |
| Salter et al[38] | Pts | 30 (0:30) | 20/30 pts on AOD adherent (n=19) declined (n=1) stopped AOD (n=10) | Interview | Framework analysis | Primary care | UK |
| **Studies in group 2: views of osteoporosis treatment (decision making)** | | | | | | | |
| Mazor et al[40] | Pts | 36 (0:36) | 15/36 pts on AOD adherent (n=15) declined (n=10) stopped (n=11) | Telephone Interview | (thematic analysis) | Primary care | USA |
| Sale et al[44] | Pts | 24 (6:15) | 14/21 pts on AOD | Telephone Interview | Phenomenological study | Hospital based fracture screening programme | Canada |
| Swart et al[42] | Mixed | Pts: 26 (4:22) HCPs: 13 (not given) | 10/26 pts on AOD adherent (n=10) declined (n=16) | Interview | Thematic analysis with elements of grounded theory | Recruited from a fracture prevention study | Netherlands |
| Scoville et al[43] | Mixed | Pt: 18 (0:18) HCP: 19 (12:7) | N/A | Videographic | (deductive checklist and descriptive) | Primary care (osteoporosis choice trial) | USA |
| Wozniak et al[41] | Pts | 12 (3:9) | 7/12 pts on AOD, adherent (n=7) stopped (n=5) | Interview | Grounded theory | Recruited from a fracture prevention trial nested in secondary care | Canada |
| **Studies in group 2: views of osteoporosis treatment (bisphosphonate side effects)** | | | | | | | |
| Sturrock et al[46] | Pts | 23 (4:19) | 13/23 pts on AOD | Interview | Grounded theory | Three regions including from secondary care | UK |
| Sturrock et al[45] | Pts | 17 (7:10) | N/A | Interview | Grounded theory | Primary care | UK |
| **Studies in group 3: non-specific osteoporosis issues** | | | | | | | |
| Ailabouni et al[30] | HCPs | 10 GPs | N/A | Interview | Constant comparison | Primary care | New Zealand |
| Sippli et al[31] | HCPs | 28 (6:22) | N/A | Interview | Content analysis | Primary care | Germany |

*Where specified. N/A not applicable.
†Text in parentheses: qualitative approach not explicitly stated.
‡AOD, antiosteoporosis drug; GP, general practitioner; HCPs, healthcare professionals; Pts, patients.

**Table 2** Quality appraisal

| Author | CASP tool question* 1 | 2 | 3 | 4 | 5 | 6 | 7 | 8 | 9 | 10 | Comments† |
|---|---|---|---|---|---|---|---|---|---|---|---|
| **Group 1: views of osteoporosis** | | | | | | | | | | | |
| Besser et al[23] | ✓ | ✓ | ✓ | p | ✓ | | ✓ | p | ✓ | Moderate | Small sample, no mention of data saturation, limited to 'psychological' factors affecting adherence (discounting other factors by omission) and some use of normative statements |
| Jaglal et al[29] | ✓ | ✓ | ✓ | ✓ | ✓ | u | ✓ | ✓ | ✓ | Moderate | Few findings relevant to our research question |
| Otmar et al[28] | ✓ | ✓ | ✓ | ✓ | ✓ | | ✓ | ✓ | ✓ | Moderate | Well conducted study, but limited findings relating to bisphosphonates |
| Sale et al[24] | ✓ | ✓ | ✓ | ✓ | ✓ | u | ✓ | ✓ | ✓ | High | |
| Sale et al[25] | ✓ | ✓ | ✓ | p | ✓ | u | ✓ | p | ✓ | Moderate | Small single site study, although data saturation reached. Language does not always appear to match approach (eg, reporting patient 'inability' to link fractures to osteoporosis suggests prior normative assumptions) |
| Weston et al[26] | ✓ | ✓ | ✓ | ✓ | ✓ | ✓ | ✓ | ✓ | ✓ | High | |
| **Group 2: views of osteoporosis treatment** | | | | | | | | | | | |
| Alami et al[35] | ✓ | ✓ | ✓ | ✓ | ✓ | | ✓ | ✓ | ✓ | High | |
| Drew et al[34] | ✓ | ✓ | ✓ | ✓ | ✓ | u | ✓ | ✓ | ✓ | High | |
| Feldstein et al[16]) | ✓ | ✓ | ✓ | ✓ | ✓ | u | ✓ | ✓ | ✓ | High | |
| Guzman-Clark et al[36] | ✓ | ✓ | ✓ | ✓ | ✓ | u | ✓ | u | ✓ | Moderate | Only partially relevant for our review given the focus on a specific population (glucocorticoid-induced osteoporosis) |
| Merle et al[32] | ✓ | ✓ | ✓ | p | ✓ | u | ✓ | u | ✓ | Moderate | Small sample (although data saturation reached) without attempt to structure to population and analysis lacks depth to answer our objective relating to bisphosphonate acceptability |
| Merle et al[33] | ✓ | ✓ | ✓ | ✓ | ✓ | u | ✓ | ✓ | ✓ | Moderate | Limited information relevant to our research question in view of general focus on osteoporosis |
| Iversen et al[39] | ✓ | ✓ | ✓ | p | ✓ | | ✓ | p | ✓ | Moderate | Single centre study, although data saturation reached, limited information on coding/analysis and no discussion of findings with relevance to wider literature |
| Lau et al[37] | ✓ | ✓ | ✓ | ✓ | ✓ | | ✓ | ✓ | ✓ | High | |
| Salter et al[38] | ✓ | ✓ | ✓ | ✓ | ✓ | | ✓ | ✓ | ✓ | High | |
| Hansen et al[27] | ✓ | ✓ | ✓ | ✓ | ✓ | ✓ | ✓ | ✓ | ✓ | High | |
| Mazor et al[40] | ✓ | ✓ | ✓ | ✓ | ✓ | u | ✓ | u | ✓ | Moderate | Good relevance, single site. Descriptive approach without critical reflexivity or discussion of prior assumptions |
| Sale et al[44] | ✓ | ✓ | ✓ | ✓ | ✓ | u | ✓ | ✓ | ✓ | High | |
| Swart et al[42] | ✓ | ✓ | ✓ | ✓ | ✓ | ✓ | ✓ | ✓ | ✓ | High | |
| Scoville et al[43] | ✓ | ✓ | ✓ | ✓ | ✓ | u | ✓ | ✓ | ✓ | Moderate | Well conducted videographic study, but data coded against deductive categories of reasons to reject treatment, so limited potential to inform our objective about acceptability |
| Wozniak et al[41] | ✓ | ✓ | ✓ | ✓ | ✓ | u | ✓ | ✓ | ✓ | High | |
| Sturrock et al[46] | ✓ | ✓ | ✓ | ✓ | ✓ | u | ✓ | ✓ | ✓ | High | |
| Sturrock et al[45] | ✓ | ✓ | ✓ | ✓ | ✓ | | ✓ | ✓ | ✓ | Moderate | Aim only partially relevant to study question |
| **Group 3: non-specific osteoporosis issues** | | | | | | | | | | | |
| Ailabouni et al[30] | ✓ | ✓ | ✓ | p | ✓ | ✓ | ✓ | ✓ | ✓ | Moderate | Relatively small (10 respondents) study, although data saturation reached. Only partially relevant for current review with brief coverage of GPs views on discontinuing bisphosphonates in light of multimorbidities |
| Sippli et al[31] | ✓ | ✓ | ✓ | ✓ | ✓ | | ✓ | ✓ | ✓ | Moderate | Limited findings related to our research question |

Continued

**Table 2** Continued

| Author | CASP tool question* | | | | | | | | | | Comments† |
|---|---|---|---|---|---|---|---|---|---|---|---|
| | 1 | 2 | 3 | 4 | 5 | 6 | 7 | 8 | 9 | 10 | |

*Critical Appraisal Skills Programme (CASP) quality assessment questions: (1) was there a clear statement of the aims of the research?; (2) is a qualitative methodology appropriate?; (3) was the research design appropriate to address the aims of the research?; (4) was the recruitment strategy appropriate to the aims of the research?; (5) was the data collected in a way that addressed the research issue?; (6) has the relationship between researcher and participants been adequately considered?; (7) have ethical issues been taken into consideration?; (8) was the data analysis sufficiently rigorous?; (9) is there a clear statement of findings?; (10) value of study and relevance to review objectives. ✓=yes, u=unsure, p=partial, blank=no.
†Comments only made for those ranked moderate or low.
GP, general practitioner.

moderate in value, this was due to methodological issues, and, for 8/13 (62%) studies this was because the focus of the paper was less relevant to our research question.

Fifteen individual subthemes were identified which mapped to the seven domains of the TFA. Key findings relating to ethicality related to conflict between bisphosphonates and participants' values and were usually discussed as part of sense making. For this reason, issues relating to 'ethicality' were considered as part of 'intervention coherence', leaving six main themes, as shown schematically in figure 2. Although it was possible to distinguish between two temporal perspectives, related to anticipated and experienced acceptability within most domains (with the exception of self-efficacy) the majority of anticipated acceptability findings related to intervention coherence.

The findings of the review are discussed later with GRADE-CERQual ratings of confidence in table 3 and illustrative key findings for each theme/subtheme shown in online supplemental file 2. Subthemes are identified in the text in italics.

### Intervention coherence (high confidence)

Both before starting, and during treatment, patients considered the perceived need or *necessity* for bisphosphonates based on their views of osteoporosis, including its seriousness and controllability, symptoms and their *perception of their own health*. Perceived need was weighed up against *concerns* about medication, including suspicion of drugs in general and specific concerns about bisphosphonate safety, by both patients and HCPs. HCPs sometimes used principles of *ethicality* to support perceptions of low necessity and their reluctance to prescribe. The *decision process* of balancing necessity against concerns, was influenced by the doctor–patient relationship and wider societal influences including friends, family and the general media. This process influenced whether HCPs reported recommending bisphosphonates. For patients, the decision process could be explicit or tacit, was revisited over time and influenced both whether they initiated treatment and subsequently adhered.

### Perceived effectiveness (high confidence)

Both patients and HCPs expressed doubt or uncertainty about the *mechanism of effectiveness* of bisphosphonates and expressed a range of treatment expectations including strengthening bone—improving bone density, preventing worsening of osteoporosis—maintaining bone density and/or total fracture prevention. Patients wanted proof or evidence of effectiveness through more structured *monitoring and follow-up*, and were disincentivised to continue treatment in the absence of evidence of perceived effectiveness.

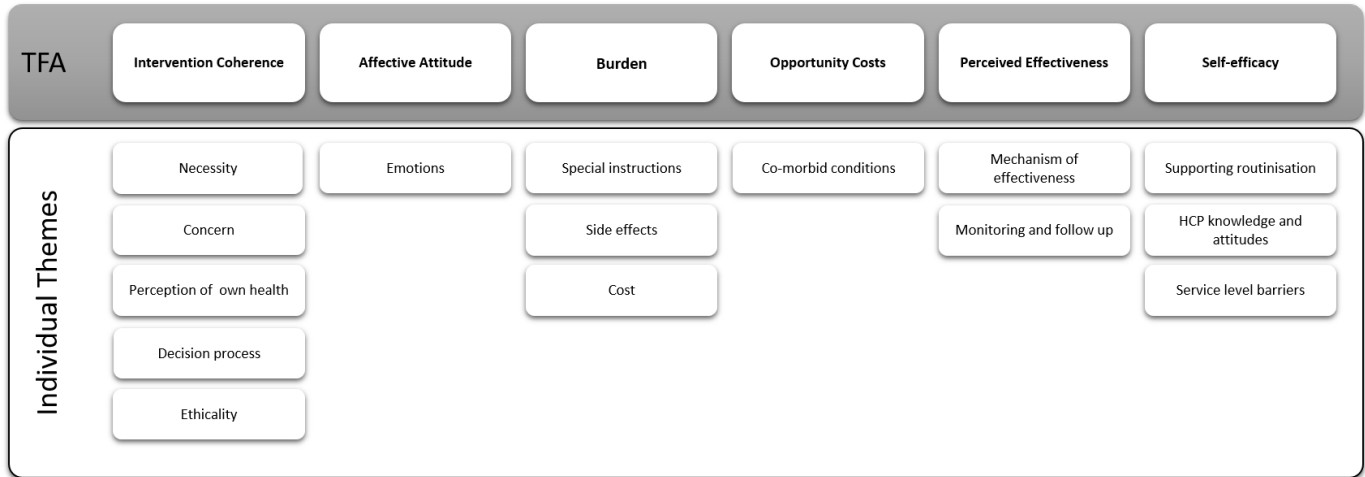

**Figure 2** Identified themes and subthemes mapped to the theoretical framework of acceptability (TFA). HCP, healthcare professional.

**Table 3** Grades of Recommendation, Assessment, Development, and Evaluation Confidence in the Evidence from Qualitative Reviews summary of qualitative findings table

| Review finding (and contributing studies) | Methodological limitations **Concerns** | Coherence | Adequacy | Relevance | CERQual confidence assessment |
|---|---|---|---|---|---|
| **Intervention coherence** Both before starting, and during treatment, patients considered the perceived need or necessity for bisphosphonates based on their views of osteoporosis, including its seriousness and controllability, symptoms and their perception of their own health. Perceived need was weighed up against concerns about medication, including suspicion of drugs in general and specific concerns about bisphosphonate safety by both patients and HCPs. HCPs sometimes used principles of ethicality to support perceptions of low necessity and their reluctance to prescribe. The decision process of balancing necessity against concerns, was influenced by the doctor–patient relationship and wider societal influences including friends, family and the general media. This process influenced whether HCPs reported recommending bisphosphonates. For patients, the decision process could be explicit or tacit, was revisited over time and influenced both whether they initiated treatment and subsequently adhered. [16 23 25–30 32 33 35–44 46] | *Minor* 12/22 papers rated moderate value due to sample size, depth of analysis or lack of reflexivity.* | *None or very minor* The finding reflects the complexity and variation of the data, and these influences on sense making are well supported by details in the underlying studies. | *None or very minor* 22 papers contributed to this finding, and although some gave little detail, in-depth insights were reported in 10 papers and information was consistent across studies. | *Minor* Spread of studies from primary and secondary care and range of countries. Uncertainties remain about sense making related patients taking intravenous bisphosphonates and influence of gender. | High |
| **Perceived effectiveness** Both patients and HCPs expressed doubt or uncertainty about the mechanism of effectiveness of bisphosphonates and expressed a range of treatment expectations including strengthening bone—improving bone density, preventing worsening of osteoporosis—maintaining bone density and/or total fracture prevention. Patients wanted proof or evidence of effectiveness through more structured monitoring and follow-up, and were disincentivised to continue treatment in the absence of evidence of perceived effectiveness. [16 23 24 29 34 35 38–40 42 43] | *Minor* 7/15 papers rated moderate value, mostly (4/7) due to limited relevant content. Methodological concerns relate to depth of analysis or lack of reflexivity.* | *None or very minor* The finding reflects the complexity and variation of the data, and these issues are supported by details in the underlying studies. | *None or very minor* 15 papers contributed to this finding. Some gave little detail, but in-depth insights were reported in six papers and information was consistent. | *Minor* Spread of studies from primary and secondary care and range of countries. Uncertainties remain about perceived effectiveness of intravenous bisphosphonates. | High |

Continued

**Table 3** Continued

| Review finding (and contributing studies) | Methodological limitations | | Adequacy | Relevance | CERQual confidence assessment |
|---|---|---|---|---|---|
| | Concerns | Coherence | | | |
| **Self-efficacy**<br>Measures to help patients integrate medication taking into daily routines (supporting routinisation), and the provision of information and support, enhanced their feeling of having control over their health and confidence to adhere to bisphosphonates. Clinician reported barriers to supporting adherence related to perceptions of their knowledge and attitudes, with several knowledge gaps and uncertainties reported, and the perception that osteoporosis was not a priority. Finally, service level barriers which impaired clinicians' self-efficacy in recommending and managing patients on bisphosphonates, included uncertainty about professional roles and responsibilities, capacity, access to intravenous drugs and communication and IT systems.<br>16 24 26 27 30–32 37 38 45 | *Minor*<br>7/15 papers rated moderate value, mostly (4/7) due to limited relevant content. Methodological concerns relate to depth of analysis or sample size.* | *None or very minor*<br>The finding reflects the complexity and variation of the data, and these issues are supported by details in the underlying studies. | *None or very minor*<br>17 papers contributed to this finding. Some gave little detail, but in-depth insights were reported in five papers and information was consistent. | *Minor*<br>Spread of studies from primary and secondary care and range of countries. Uncertainties remain about self-efficacy relating to intravenous bisphosphonates. | High |
| **Affective attitudes**<br>The emotions elicited by bisphosphonates were closely related to intervention coherence. Bisphosphonates were associated predominantly with negative emotions of fear (of side effects) and annoyance (with special instructions); however, positive emotions of reassurance and hope were noted in two studies, linked to the anticipated protection that bisphosphonates could incur.<br>16 23 26 27 35 37 38 40 | *Minor*<br>2/8 papers rated moderate value due to depth of analysis or lack of reflexivity.* | *None or very minor*<br>The finding reflects the data, supported by details in the underlying studies. | *Moderate*<br>Reports of affective attitude were mostly descriptive with little depth. | *Moderate*<br>Uncertainties remain about affective attitudes to injectable bisphosphonates received in hospital. | Moderate |
| **Burden**<br>The burden or effort of oral bisphosphonates was described mostly relating to the special instructions to take oral bisphosphonates or experienced side effects, although costs incurred were also a potential source of burden.<br>16 23 26 27 32 37–39 42 43 46 | *Minor*<br>4/11 papers rated moderate value due to sample size, depth of analysis.* | *None or very minor*<br>The finding reflects the data, and these aspects of burden are supported by details in the underlying studies. | *Moderate*<br>Reports mostly descriptive with little depth and a possible focus on presence of burden (side effects) rather than absence. | *Moderate*<br>Uncertainties remain about burden of indirect costs (travel, dental checks) and burden due to intravenous bisphosphonates. | Moderate |
| **Opportunity costs**<br>Circumstances where competing priorities challenged adherence or initiation of bisphosphonates were described relating to comorbid conditions. The presence of comorbid conditions were described as resulting in less time to support discussion about bisphosphonates in consultations and, result in recommendation of, and adherence to, bisphosphonates being given relative low priority.<br>16 27 29 32 33 38 41 42 44–46 | *None or very minor*<br>4/11 papers rated moderate value, but this was mostly relevant content rather than methodological concerns. | *Moderate*<br>No discussion of the alternative explanation that having comorbid conditions may facilitate bisphosphonate acceptability. | *Moderate*<br>Reports were limited, lacked depth and three papers contained little content relevant to the research question. | *Moderate*<br>No information about values, benefits that have to be given up to partake in intravenous bisphosphonates, which are likely to be different and likely limited sampling of patients with complex health needs. | Low |

*Concerns considered minor because of the methodological strength of the other papers in this domain, and low likelihood that reflexivity would affect finding.

CERQual, Confidence in the Evidence from Qualitative Reviews.

### Self-efficacy (high confidence)

Measures to help patients integrate medication taking into daily routines (*supporting routinisation*), and the provision of information and support, enhanced their feeling of having control over their health and confidence to adhere to bisphosphonates. Clinician reported barriers to supporting adherence related to perceptions of their *knowledge and attitudes,* with several knowledge gaps and uncertainties reported, and the perception that osteoporosis was not a priority. Finally, s*ervice level barriers* which impaired clinicians' self-efficacy in recommending and managing patients on bisphosphonates, included uncertainty about professional roles and responsibilities, capacity, access to intravenous drugs and communication and IT systems.

### Affective attitudes (moderate confidence)

The *emotions* elicited by bisphosphonates were closely related to intervention coherence. Bisphosphonates were associated predominantly with negative emotions of fear (of side effects) and annoyance (with special instructions); however, positive emotions of reassurance and hope were noted in two studies, linked to the anticipated protection that bisphosphonates could incur.

### Burden (moderate confidence)

The burden or effort of oral bisphosphonates was described mostly relating to the *special instructions* to take oral bisphosphonates or experienced *side effects*, although *costs* incurred were also a potential source of burden. Only one study included the experience of a patient on an intravenous bisphosphonate, this patient described low treatment burden as she only had to go once a year, and felt no side effects.[31]

### Opportunity costs (low confidence)

There were few descriptions of 'benefits, profits, or values' being given up to take bisphosphonates. However, circumstances where competing priorities challenged adherence or initiation of bisphosphonates were described relating to *comorbid conditions*. The presence of comorbid conditions was described as resulting in less time to support discussion about bisphosphonates in consultations and, result in recommendation of, and adherence to, bisphosphonates being given relative low priority.

### DISCUSSION

This systematic review has used the lens of acceptability to understand perceptions of bisphosphonates and the problem of poor adherence. We have identified, with high confidence, how patients and HCPs make sense (coherence) of bisphosphonates by balancing perceptions of need against concerns, how uncertainty prevails about perceived effectiveness of bisphosphonates and how a number of individual and service factors have potential to increase self-efficacy in recommending and adhering to bisphosphonates. We identified with moderate confidence, that bisphosphonate taking induces fear, but has the potential to engender reassurance, and that both the side effects and special instructions for taking oral bisphosphonates can be a source of treatment burden. Finally, we identified with low confidence that multimorbidity plays a role in people's perception of bisphosphonate acceptability.

To our knowledge, this is the first use of the TFA, originally developed to evaluate acceptability of complex interventions, to evaluate the acceptability of medication. We explored the utility of the TFA from two perspectives, as an explanatory model for both patient and clinician acceptability and engagement. The TFA was useful for understanding and combining patient and clinician viewpoints; however, there was considerable overlap between domains; perceived efficacy, affective attitudes and self-efficacy beliefs are all likely to impinge on sense-making, or intervention coherence. The TFA alone does not provide a comprehensive framework for understanding patient acceptability or engagement with medicines, and of course it was not intended to do so. The sense-making aspect of the framework appeared pivotal, and the explanatory value of the framework was enhanced by the incorporation of the NCF to operationalise key engagement related beliefs. In the context of bisphosphonates, concern and associated fears predominate among patients, and perceived need may be underestimated if the consequences of osteoporosis and fragility fractures are not explained. In our findings, sense making was dynamic. Patients re-evaluated perceptions of bisphosphonates over time, expressing uncertainty relating to what represents successful treatment and citing perceived lack of effectiveness being cited as reason to discontinue. This is likely to be a particular problem for bisphosphonates, as opposed to other drugs commonly taken for prevention such as statins and antihypertensive, where measures of feedback and effectiveness are more readily available.

The UK National Institute for Health and Care Excellence (NICE) guidelines for medicines adherence emphasises the need to take into account perceptions (eg, necessity beliefs and concerns) and practicalities (eg, capability and resources) that will affect individuals' motivation and ability to start and continue with treatment.[47] However, interventions designed to improve bisphosphonate adherence are often designed to 'educate' or persuade the patient of importance and are often not targeted to eliciting or addressing health beliefs, or informed by underpinning mechanisms of change.[3] There is therefore a need to ensure that any further design of interventions—to promote bisphosphonate adherence—draws on more comprehensive theoretical models of patient engagement with health conditions and medicines such as the Extended Common Sense Model.[48] This model situates individual's perceptions about drugs, and practical issues related to capability, in the context of illness and treatment representations.

Specifically, our findings suggest a need for clinicians to support patients to understand the need for treatment, to

allay concerns where possible and to define what constitutes successful bisphosphonate treatment. Furthermore, clinicians need to support patients evaluate the advantages and disadvantages over time, given the dynamic nature of these decision processes.[48]

It is clear from our findings that clinicians also have necessity-concern dilemmas relating to bisphosphonates. A number of studies reported clinicians themselves perceiving low patient need, high concerns and perceptions treatment was not practical. This is perhaps in contrast with a previous quantitative study in asthma which demonstrated that clinicians held stronger positive beliefs about medicines than patients.[49] It is unclear to what extent the perceptions in our findings were generalisations or applied in specific circumstances, or to what extent these views were negotiated on an individual basis in discussion with patients. Problems may arise in the consultation if clinicians assume patients share their views and then may be less likely to explore patient perceptions of need or concerns. Furthermore, the limitations of interviewing HCPs are well documented; the accounts presented in an interview may not represent clinician underlying beliefs or behaviours meaning that observational methods may be more appropriate to fully understand clinical decisional making.[50] Given the clinician has a pivotal role in sense making, interventions are also likely needed to address clinician knowledge, attitudes and beliefs. By including the views of clinicians and managers we have also identified a range of service level barriers to promoting bisphosphonate adherence relating to lack of clarity about professional roles, both across primary and secondary care, and within primary care, use of IT systems and access to intravenous treatments.

A strength of this review is the comprehensive search, use of underpinning theoretical framework, the inclusion of clinician views in addition to patients, and the use of the GRADE-CERQual to give confidence in our findings which has facilitated a clear identification of where further research is needed. Areas where we have identified moderate or low confidence are in need of further research and specifically relate to the influence of multimorbidity on sense making, burden and self-efficacy in bisphosphonate users, the extent to which intravenous bisphosphonates may overcome issues related to treatment burden and self-efficacy, and the impact of bisphosphonates on affective attitudes and emotions. Furthermore, we have identified gaps in our understanding of how clinicians make decisions in practice, and how views of bisphosphonates may be influenced by gender. Given that many osteoporosis drugs have a different evidence base and licensing arrangements in men this is an area in need of further study.

The main limitation of this study relates to the lack of clarity in many of the included studies in the results sections about which osteoporosis treatments or bisphosphonates were being referred to, meaning that in some cases we may have over-interpreted findings relating to bisphosphonates that were about other osteoporosis drugs. However, all of our review findings were identified from comparison of data from several studies, and as bisphosphonates represent the mainstay of osteoporosis treatment, we consider that over-interpretation is unlikely. As there was frequently little detail about medication participants were taking or referring to, it is also possible that we have missed relevant studies. The views of men were under-represented; although 8/18 studies included men, men represented less than 20% of the total patient population in the included studies. It is important for future studies to include males and specific populations such as those with glucocorticoid-induced osteoporosis who are likely to have different experiences and needs.[51] Only two studies reported the views of managers but unfortunately neither of these studies distinguished professional roles in the presentation of results, so a further need exists to explore perceptions of this group, and perceptions of payors and academics. Finally, although the population from which each study sampled was reasonably well described, it was not always possible to appreciate if the setting was primary or secondary care; the majority of studies appeared to recruit from primary care which may explain the lack of findings related to intravenous bisphosphonates and limit the transferability of our findings to non-primary care settings.

## CONCLUSIONS

In summary, using the lens of acceptability, we have identified the factors that influence how patients and clinicians make sense of bisphosphonates, described the experience of bisphosphonate taking in terms of burden and factors that both facilitate and hinder confidence in taking, and prescribing and monitoring bisphosphonates. Our findings demonstrate the need for a theoretically informed, whole-system approach' to enable clinicians and patients to get the best from bisphosphonate treatment. Patients need comprehensive support that takes account of the perceptions (eg, treatment necessity beliefs and concerns) and practicalities (eg, capability and resources) that influence their motivation and ability to start and continue with treatment. Clinicians need to moderate patient expectations and clarify what constitutes bisphosphonate treatment success. Finally, further research is needed to explore perspectives of managers, patients receiving intravenous bisphosphonates, men receiving bisphosphonates and the use of bisphosphonates in the context of multimorbidity.

**Author affiliations**
[1]School of Medicine, Keele University, Keele, UK
[2]Haywood Academic Rheumatology Centre, Haywood Hospital, Stoke-on-Trent, UK
[3]Centre for Health Innovation, Leadership and Learning, University of Nottingham, Nottingham, UK
[4]Operations and Clinical Practice, Royal Osteoporosis Society, Bath, UK
[5]Faculty of Medicine, University of Malaya, Kuala Lumpur, Malaysia
[6]Centre for Endocrinology Diabetes and Metabolism, University of Birmingham, Birmingham, UK
[7]Faculty of Medicine & Health Sciences, University of Nottingham, Nottingham, UK

[8]Division of Epidemiology and Public Health, University of Nottingham, Nottingham, UK

[9]School of Pharmacy, University College London, London, UK

[10]Department of Geriatric Medicine, Nottingham University Hospitals NHS Trust, Nottingham, UK

**Contributors** Conceptualisation: ZP, SB, EC, AD, TO, NG, JL-B, TL, OS. Protocol: ZP, EC, NC, SB, JL-B, TL, TO, OS, NG, AD. Search implementation: ZP, EC, NC, JW, FC-M. Data extraction and quality: ZP, EC, NC, JW, FC-M, SB. Synthesis: ZB, SB, EC, FC-M, RH, CJ. Writing- original draft: ZP, FC-M. Writing-review and editing: ZP, SB, AD, TO, NG, JL-B, CJ, TL, OS, FC-M, JW, NC, NH, EC.

**Funding** This study is funded by the National Institute for Health Research (NIHR), [HTA NIHR127550]. ZP is funded by the NIHR, Clinician Scientist Award (CS-2018-18-ST2-010)/NIHR Academy. CJ is part funded by the NIHR Applied Research Collaboration West Midlands. The views expressed are those of the author(s) and not necessarily those of the National Health Service, the NIHR, or the Department of Health and Social Care.

**Competing interests** None declared.

**Patient consent for publication** Not required.

**Provenance and peer review** Not commissioned; externally peer reviewed.

**Data availability statement** All data relevant to the study are included in the article or uploaded as supplementary information.

**ORCID iDs**
Fay Crawford-Manning http://orcid.org/0000-0002-9768-1695
Elizabeth Cottrell http://orcid.org/0000-0002-5757-1854

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
