## [Reviewer comments · BMJ Open]

ARTICLE DETAILS

TITLE (PROVISIONAL)	Acceptability of bisphosphonates among patients, clinicians and managers: a systematic review and framework synthesis.
AUTHORS	Paskins, Zoe; Crawford-Manning, Fay; Cottrell, Elizabeth; Corp, Nadia; Wright, Jenny; Jinks, Clare; Bishop, Simon; Doyle, Alison; Ong, Terence; GITTOES, NEIL; Leonardi-Bee, Jo; Langley, Tessa; Horne, Robert; Sahota, Opinder

VERSION 1 – REVIEW

REVIEWER	Dr Zahra Cheraghi Hamadan University of Medical Science
REVIEW RETURNED	21-Jun-2020

GENERAL COMMENTS	Dear editor This systematic review is about the "Acceptability of bisphosphonates among patients, clinicians and managers: a systematic review and framework synthesis". The paper is well written. The methodology was good and vigorous. Some revisions as follow as: Methods: 1. What kind of qualitative studies (grounded theory? Conversation analysis? Or content analysis...) included in this review? in table 1 this variable should be added?2. Searching: Present full electronic search strategy for at least one database, including any limits used, such that it could be repeated Quality appraisal: CASP tool for qualitative studies has 10 items, the authors should be explaining what questions they were assessed? It sounds due to many questions in their appraisal, the Quality appraisal assessed very Strictly! Results: 1. Present results of any assessment of risk of bias across studies. What percentage of studies was of good, moderate or weak quality? Totally the generalizability of qualitative studies is not sufficient, explain this in more detail.
---

REVIEWER	Beauvais Catherine Hôpital Saint Antoine. Rheumatology Department Sorbonne université .Assistance Publique hopitaux de Paris France
REVIEW RETURNED	08-Jul-2020

GENERAL COMMENTS	bmjopen-2020-040634 - Acceptability of bisphosphonates among patients, clinicians and managers: a systematic review and framework synthesis. Thank you for giving me the opportunity to review the manuscript. Acceptability is a major issue in osteoporosis management and qualitative studies are of great interest to catch the patients' unbiased perspectives and experiences. The interests of the study in OP are as follows  - Special attention to bisphosphonates, the most frequently prescribed in osteoporosis. - the methodology using the theoretical Framework of Acceptability - inclusion of patients, health care professionals and managers involved the prescribing of bisphosphonates This review did not show new determinants for BP acceptability but shows relevant insights to understand the general reluctance of patients and HCP towards BPs , with a mirrored perception between patients and prescribers. The conclusions tends towards a holistic management of OP. Comments on the text below.  1. In the methods, make it clear that the CASP tool was used to rate the studies quality also in relation with the objectives in the study to explore perspectives on bisphosphonates. 2. Mention that the qualitative studies have focused on post-menopausal women whereas other populations with osteoporosis such as men have been rarely considered. Patients with glucocorticosteroid-induced osteoporosis were not included in qualitative studies although acceptability is a major issue in these populations. (mention in the discussion the study by Beauvais et al sept 2019 CTI). 3. The study does not explain the specificity of intravenous BPs 4. The authors conclude that the theoretical Framework of Acceptability doesn't provide a comprehensive understanding of patients adherence and must be complemented with the widely explored necessity and concerns framework. Can the authors comment how to integrate the health beliefs model within these theoretical models ? Which models should most appropriate models for future research ?
--

VERSION 1 – AUTHOR RESPONSE

Reviewer 1	
Methods 1. What kind of qualitative studies (grounded theory? Conversation analysis? Or content analysis...) included in this review? in table 1 this variable should be added?	Thank you. We have now added this important detail in Table 1 with a column entitled 'Qualitative approach or analysis method'.

2. Searching: Present full electronic search strategy for at least one database, including any limits used, such that it could be repeated	Thank you. A full MEDLINE search was included in Supplementary File 2.
Quality appraisal: CASP tool for qualitative studies has 10 items, the authors should be explaining what questions they were assessed? It sounds due to many questions in their appraisal, the Quality appraisal assessed very Strictly!	Thank you. Apologies for omitting to include this. This is now included in Supplementary File 2.
Results: 1. Present results of any assessment of the risk of bias across studies. What percentage of studies were of good, moderate, or weak quality?	Thank you. We find that 'bias' is not an appropriate term for qualitative studies. Quality was assessed using the CASP tool. We had already summarised the methodological quality in the statement relating to CASP tool results, but have added additional words to make clearer the CASP tool was used both to rate quality and to rate the degree to which included papers met our research objectives For 5/13 (38%) studies scored as moderate in value, this was due to methodological issues, and, for 8/13 (62%) studies this was because the focus of the paper was less relevant to our research question.
Totally the generalizability of qualitative studies is not sufficient, explain this in more detail. -	Thank you. We agree this is important to comment on. We prefer to use the term transferability when referring to qualitative research findings as qualitative research does not seek to be generalizable. We have added the following sentence to the paragraph on limitations in the discussion which we hope has address this comment: Finally, although the population from which each study sampled was reasonably well described, it was not always possible to appreciate if the setting was primary or secondary care; the majority of studies appeared to recruit from primary care which may explain the lack of findings related to intravenous bisphosphonates and limit the transferability of our findings to non-primary care settings.

Reviewer 2	
1. In the methods, make it clear that the CASP tool was used to rate the studies quality also in relation with the objectives in the study to explore perspectives on bisphosphonates.	Thank you for highlighting this. We have clarified as follows: The final section question was an assessment based on the overall quality of the paper; this was informed by response to including the previous items (indicating methodological quality) and by the study's perceived its relevance of the study to the review objectives, this and was rated as "high", "moderate" or "low".
2. Mention that the qualitative studies have focused on post-menopausal women whereas other populations with osteoporosis such as men have been rarely considered. Patients with glucocorticosteroid-induced osteoporosis were not included in qualitative studies although acceptability is a major issue in these populations. (mention in the discussion the study by Beauvais et al sept 2019 CTI).	
3. The study does not explain the specificity of intravenous BPs	Thank you we agree with this important limitation which we had already highlighted as an area for further research. We have also added this statement to the limitations paragraph in the discussion: Finally, although the population from which each study sampled was reasonably well described, it was not always possible to appreciate if the setting was primary or secondary care; the majority of studies appeared to recruit from primary care which may explain the lack of findings related to intravenous bisphosphonates and limit the transferability of our findings.
4. The authors conclude that the theoretical Framework of Acceptability doesn't provide a comprehensive understanding of patients adherence and must be complemented with the widely explored necessity and concerns framework. Can the authors comment how to integrate the health beliefs model within these theoretical models ? Which models should most appropriate models for future research ?	Thank you for this important comment. We agree that suggesting an alternative theoretical model is important. We had already suggested the Extended Common Sense Model (ECSM) as an alternative, rather than the Health Beliefs model, as the ECSM is a psychological health behaviour change model developed to explain and predict health-related behaviours relating to drug taking and adherence specifically. We have added in an additional sentence to explain the ECSM: There is therefore a need to ensure that any further design of interventions - to promote bisphosphonate adherence - draws on more

	comprehensive theoretical models of patient engagement with health conditions and medicines such as the Extended Common Sense Model.[48] This model situates individual's perceptions about drugs, and practical issues related to capability, in the context of illness and treatment representations.
--	---

VERSION 2 – REVIEW

REVIEWER	Dr. Zahra Cheraghi Iran. Hamadan University of Medical Sciences.
REVIEW RETURNED	06-Sep-2020

GENERAL COMMENTS	Dear Editor The edited files were checked, all changes were done correctly and my final decision is ACCEPT. Yours sincerely Cheraghi
---

REVIEWER	Beauvais Catherine Hôpital Saint Antoine. Rheumatology Department Sorbonne université .Assistance Publique hopitaux de Paris France
REVIEW RETURNED	21-Sep-2020

GENERAL COMMENTS	Thank you for having revised the manuscript et considered the reviewers' comments.
--